# Characterizing A21: Natural Cyanobacteria-Based Consortium with Potential for Steroid Bioremediation in Wastewater Treatment

**DOI:** 10.3390/ijms252313018

**Published:** 2024-12-04

**Authors:** Govinda Guevara, Jamileth Stefania Espinoza Solorzano, Marta Vargas Ramírez, Andrada Rusu, Juana María Navarro Llorens

**Affiliations:** 1Department of Biochemistry and Molecular Biology, Universidad Complutense de Madrid, c/Jose Antonio Novais 12, 28040 Madrid, Spain; jamiespi@ucm.es (J.S.E.S.); andradar@ucm.es (A.R.); 2Department of Genetics, Physiology and Microbiology, Universidad Complutense de Madrid, c/Jose Antonio Novais 12, 28040 Madrid, Spain; marvar05@ucm.es

**Keywords:** consortium, *Cyanobium*, wastewater, steroids, bioremediation

## Abstract

Microalga–bacteria consortia are increasingly recognized for their effectiveness in wastewater treatment, leveraging the metabolic synergy between microalgae and bacteria to enhance nutrient removal and overall treatment efficiency. These systems offer a sustainable approach to addressing pollutants such as nitrogen and phosphorus. However, their potential in removing specific contaminants like steroid hormones is less explored. In this study, a natural microbial consortium, A21, has been characterized and isolated from primary sewage treatment in Madrid and its potential for bioremediation of steroid hormone effluents has been evaluated. The A21 consortium includes Alphaproteobacteria genera *Sphingopyxis* and *Pseudorhizobium* and the Cyanobacterium *Cyanobium*. *Sphingopyxis* (31.78%) is known for biodegradation, while *Pseudorhizobium* (15.68%) exhibits detoxification abilities. *Cyanobium* (14.2%) may contribute to nutrient uptake and oxygen production. The effects of pH, nitrogen sources, and Sodium chloride concentrations on growth were evaluated. The optimal growth conditions were determined to be a pH range of 7 to 9, a salt concentration below 0.1 M, and the presence of a nitrogen source. The consortium also demonstrated effective growth across various types of wastewaters (primary, secondary, and tertiary treatment effluents). Additionally, A21 exhibited the ability to grow in the presence of steroids and transform them into other compounds, such as converting androstenedione (AD) into androsta-1,4-diene-3,17-dione (ADD) and β-estradiol into estrone.

## 1. Introduction

Natural microbial consortia can be defined as an ecosystem that combines different microorganisms, establishing different interactions (e.g., cooperation) among them [1], with community composition being a dynamic process [2]. A microbial consortium is composed of at least two different microbial populations, and this combination or association of microorganisms results in different, usually more powerful, metabolic capabilities than those of single strains [1]. Consortia display several advantages over monocultures, such as the following: (i) they are more robust to environmental challenges, (ii) they exhibit reduced metabolic burden due to a division of labor and exchange of resources, (iii) they possess expanded metabolic capabilities relative to monocultures, and (iv) they can communicate (chemically or physically) between species [3,4,5]. Moreover, natural consortia have been used in the production of foods, the recycling of micronutrients, and maintaining the health of humans, animals, and plants [3,6]. They are also being proposed for developing sustainable energy sources, substitutes for conventional fuels [7], bioremediation of metalloid polluted environments [8], and, in general, for soil pollutant removal. Microalgae and cyanobacteria are ubiquitous and grow in a wide variety of aquatic and terrestrial habitats, even in extreme environments [9]. Their presence in a consortium is beneficial as they can obtain energy through oxygenic photosynthesis. The treatment of wastewater using microalgae–bacteria consortia represents a promising strategy for the removal of nutrients such as phosphorus and nitrogen, which contribute to eutrophication and environmental degradation, supporting a sustainable approach toward a circular bioeconomy [10]. Nevertheless, micropollutants have received limited attention in studies on microalgal-based treatment systems [11].

The increasing consumption of pharmaceuticals in recent years has emerged as an environmental concern due to the accumulation of these compounds in aquatic ecosystems, where they can exert adverse effects on non-target species [12]. The bioaccumulation of these emerging micropollutants in organisms has been shown to disrupt metabolic and physiological functions [13,14]. Furthermore, the recalcitrant nature of these pollutants impedes their degradation, allowing them to persist in the environment for prolonged periods [15].

Steroids are a significant class of these micropollutants. They are tetracyclic triterpenoid lipids with a core structure consisting of 17 carbon atoms arranged in three cyclohexane rings and one cyclopentane ring. Their chemical structure makes them especially recalcitrant to degradation [16]. These compounds are widely distributed in nature and can contaminate water resources, threatening ecosystems by acting as endocrine disruptors, even at low concentrations [17]. Recent advances in environmental analytical chemistry have enabled the detection of these hormones in aquatic systems even at sub ng/L quantities, revealing them as widespread contaminants that cause environmental toxicity and adverse health effects on humans and aquatic life [18,19]. In fact, the high persistence rate of some of them allows them to be used as reference biomarkers in some environmental pollution analyses [17]. Despite the efficiency of some wastewater treatment techniques, there is still a need for more sustainable and effective methods to remove these hormones [14]. In recent years, microalgae–bacteria consortia have emerged as a promising biological tool for the remediation of wastewaters [20]. While numerous studies have focused on nutrient removal (e.g., phosphorus and nitrogen) using natural or synthetic microalgae–bacteria consortia [10,21], research targeting steroid removal is comparatively limited. Most studies have utilized consortia involving microalgal species like *Scenedesmus* sp. and *Chlorella* sp. [11,22]. These consortia have demonstrated significant potential in removing emerging contaminants such as 17β-estradiol and other estrogens from wastewater, achieving high removal efficiencies even under varying environmental conditions. These studies also highlight the role of co-cultured systems and microbial diversity in enhancing the degradation process [11,22,23]. Moreover, research has shown that algae and duckweed can accelerate estrogen removal by absorbing these compounds, which are then degraded by associated microorganisms [24]. However, there is a notable gap in the exploration of cyanobacteria–bacteria consortia for steroid removal, despite their potential for effective bioremediation in wastewater treatment systems. The effectiveness of these consortia in treating a range of contaminants, including synthetic hormones and pharmaceutical drugs, underscores the need for further investigation into diverse consortia, particularly involving cyanobacteria, to broaden the scope of wastewater treatment applications. Therefore, it is important to highlight the importance of diverse microbial systems in addressing environmental contaminants. In our work, the natural microbial consortium designated as A21 was studied for its potential in the bioremediation of steroid hormone effluents, further expanding the scope of wastewater treatment applications.

## 2. Results

### 2.1. Isolation and Identification of a Microbial Consortium from Wastewaters

Secondary treatment wastewater from Madrid sewage was used for a preliminary screening on BG11, BG13, and UTEX plates, searching for cultured cyanobacteria, axenic or in consortium. After several streaking steps, single colonies were analyzed. This work describes the isolate number A21 grown on BG11 medium.

For a first characterization approach, 16S primers specific for cyanobacteria (Appendix A) were used for PCR amplification of the DNA extracted from the A21 sample followed by subsequent sequencing. Analysis of the sequences using BLAST classified the strain as belonging to the genus *Cyanobium*, within the order Synechococcales, showing 100% identity with *Cyanobium* sp. CHAB 6568, accession number MT488300.1. No traces of eukaryotic strains were probed using specific 18S rRNA primers (Appendix A). Then, a metagenomic study was performed on the A21 isolate. A generic bacterial 16S rRNA gene-based amplicon was used to obtain the bacterial community structure (Appendix A). The data confirmed a composition of organisms falling mostly in the classes Alphaproteobacteria and Cyanobacteria (Table 1). In addition, *Sphingopyxis* was the most abundant (32%) followed by *Pseudorhizobium* (16%) and *Cyanobium* (14%).

### 2.2. Growth Characterization of A21 Isolate

For morphological characterization, the A21 isolate was analyzed both under a magnifying glass and under a microscope (Figure 1A,B). Under the magnifying glass and in the conditions described in Section 4.1, only the cyanobacteria could be observed. Under the microscope, it has a shape between oval and cylindrical, with a pale green color. Solitary cells were found, but most of them in pairs, which happens before they divide by binary fission [25].

For growth characterization, measurement was carried out at 750 nm, a wavelength that that enables the observation of microbial growth while minimizing interference from pigment absorption, which typically occurs at wavelengths below 700 nm. The A21 growth curve on BG11 was determined for 15 days in triplicate by measuring the OD_750nm_ and counting the number of cells (Figure 1C,D). The latency period observed was 3 days, with a doubling time of 3.11 ± 0.07 days under the tested conditions (Appendix A). The maximum OD_750nm_ obtained was a value of 4.6 ± 0.6 after eleven days of growth (Figure 1C).

In order to assess how different compounds might affect A21 growth, several conditions were examined: (i) varying salt concentrations to determine if the consortium could survive in marine waters, which are more abundant and cheaper than freshwater; (ii) the effect of nitrogen sources, including urea, to study alternative growth conditions; and (iii) the impact of external sugars on the consortium that could potentially interfere with metabolite exchange among microorganisms (Figure 2 and Appendix A).

The A21 isolate was unable to grow on media with high salinity, only tolerating 10 days on BG11 medium with a salt concentration of 0.1 M, reaching an average OD_750_ value of 1.1 ± 0.6, 31% of the value of the control (Figure 2A). A comparison of growth parameters between the control and 0.1M NaCl conditions revealed no significant differences, except in the growth yield parameter, which measures biomass. Under 0.1 M NaCl, the growth yield was nearly half that of the control (2 vs. 4.7) (Appendix A).

The growth of A21 isolate was studied in media with different nitrogen sources and the OD_750nm_ measurements taken after 10 days (Appendix A) proved that it could barely survive without a nitrogen source. This suggests that the consortium does not possess the ability to fix nitrogen. On the other hand, the presence of urea, even at 8 mM, in the medium proved to be toxic to the isolate (Figure 2B).

The heterotrophic growth study of the A21 isolate was conducted over 30 days with various sugars (glucose, fructose, mannose, sucrose, or maltose). A21 was unable to grow on any of these sugars, whether monosaccharide or disaccharide. Thus, it could be concluded that this consortium cannot use these alternative carbon sources (Appendix A).

One of the basic parameters influencing degradation processes is pH and, therefore, the optimum pH for A21 growth was evaluated (Figure 2). This is particularly relevant, as the pH of sewage could vary depending on the substances discharged. The A21 isolate exhibited optimal growth at pH 7 to pH 9, showing greater sensitivity to acidic pH values than to basic ones (Figure 2C). Statistical analysis indicates that the most significant differences were observed in the growth yield parameter, which decreased from 5 under control conditions to 1 when grown at pH 11 (Appendix A).

### 2.3. Triparental Mating Assays

To evaluate the potential for genetic manipulation of the cyanobacteria present in the consortium, the first critical step involved systematically determining their antibiotic sensitivity profiles. This preliminary assessment is essential for identifying appropriate selection markers for future genetic engineering efforts and ensuring the success of subsequent transformations. The antibiogram revealed that A21 was sensitive to most of the antibiotics tested even at low concentrations (Section 4.7.1), with the exception of gentamicin up to 8 μg/mL. Based on these results, Km was chosen as the selection marker for the triparental conjugation. To assess whether any cell of the consortium could be transformed, triparental conjugation studies were carried out. Despite using a high concentration of the isolate A21 (OD_750nm_ = 20), no transformation was observed with any of the cargo plasmids tested, including pSEVA 221, 231, 241, and 251, which have different origins of replication, nor with plasmids containing the CRISPR/Cpf1 cassette (pSL2680 and pSEVA251-Cpf1).

One major challenge in transformation is the presence of restriction systems. If one of the DNA inserts contains recognition sequences for any of these restriction systems, the efficiency of the conjugation is drastically reduced. To alleviate this problem, the use of helper plasmids that encode methylases that protect against these restriction systems is usually used [26]. In our case, despite using the helper plasmid pRL623, which encodes the methylases M.*Ava*I, M.*Eco*47II, and M.*Eco*T22I, no transformed cultures were obtained.

### 2.4. Growth in Wastewater and Analysis of Steroid Biotransformation by Thin-Layer Chromatography (TLC)

Firstly, the potential for growth in wastewater was investigated, evaluating the survival and proliferation of the A21 consortium across various wastewater samples (Figure 3). The A21 consortium was able to grow in all tested wastewater types, including those from primary, secondary, and tertiary treatment processes. However, the limited nutrient availability in the tertiary-treated samples appeared to hinder the consortium’s growth.

To assess the potential of A21 for steroid bioremediation, its capacity for biotransformation or biodegradation of the steroids cholesterol (CHO), 4-androstene-3,17-dione (AD), and β-estradiol was evaluated. Preliminary results confirmed that A21 was able to grow in the presence of these steroids (Appendix A). The growth medium was analyzed by TLC (Figure 4). No intermediates were detected in the presence of CHO; however, for AD, a band corresponding to 1,4-androstadien-3,17-dione (ADD) appeared after 10 days and intensified over the next 25 days, at which point the AD band had nearly disappeared. This suggests that A21 can biotransform AD into ADD. Similarly, after 20 days of growth on β-estradiol, an intermediate corresponding to estrone (E1) was observed (Figure 4).

## 3. Discussion

This study aims to characterize the natural cyanobacteria-based consortium A21, isolated from wastewater, with a focus on its microbial composition and potential applications in wastewater treatment.

The A21 consortium is primarily composed of several Alphaproteobacteria strains and a single Cyanobacteria. A member of the *Sphingopyxis* genus represents the largest proportion of the consortium, accounting for 31.78%. *Sphingopyxis* spp. have been identified in diverse environments, including contaminated soils, activated sludge, marine ecosystems, and freshwater habitats. This genus is well known for its biodegradation capabilities, metabolic versatility, and remarkable environmental adaptability [27]. These bacteria are Gram-negative, non-sporulating, rod-shaped aerobes that grow optimally at temperatures ranging from 25 to 30 °C in Luria–Bertani (LB) medium.

The second most abundant genus in the consortium is *Pseudorhizobium*, constituting 15.68% of the community. This genus is notably for its diversification through the acquisition of unique metabolic pathways, which confer specific detoxification mechanisms. Members of *Pseudorhizobium* are particularly remarkable for their extreme specialization towards chemolithoautotrophy and their resistance to toxic heavy metals [28]. This strain demonstrates adaptability to extreme environments through various detoxification and nutritional strategies, enabling survival in harsh, nutrient-poor conditions [28]. It may also possess aromatic compound degradation pathways [29].

The third most abundant genus in the consortium, constituting 14% of the community, belongs to a different class, Cyanobacteria, which are Gram-negative bacteria capable of obtaining their energy through oxygenic photosynthesis. Cyanobacteria thrive in diverse aquatic and terrestrial habitats, including extreme environments, and are among the most significant global contributors to carbon dioxide (CO_2_) and nitrogen fixation [9,30,31]. Within this group, Picocyanobacteria play a pivotal role as primary producers in aquatic ecosystems, with *Cyanobium* and *Synechococcus* being the predominant genera in freshwater environments [31]. Picocyanobacteria exhibit phenotypic plasticity, appearing either as single cells or in aggregates [32]. In the A21 consortium, *Cyanobium* is observed as single cells (Figure 1B). This is consistent with reports that these organisms form colonies under specific stress conditions, such as predation, which are absent in the A21 consortium [33].

Picocyanobacteria have evolved various strategies to adapt to diverse environmental conditions, among them changing the composition of their accessory pigments such as phycoerythrin and phycocyanin, which enable them to efficiently capture light [30]. This family is a frequent component of microbial consortia and, in fact, it has been recently reported that picocyanobacterial–bacterial interactions can sustain cyanobacterial blooms in nutrient-limited aquatic environments [34]. A stable and tight co-occurrence pattern has been observed between dominant cyanobacteria (*Synechococcus* and *Cyanobium*, among others) and certain heterotrophic bacteria (Proteobacteria, Actinobacteria, and Bacteroidetes). These interactions contribute to the remineralization of organic matter for cyanobacteria utilization. *Cyanobium* is also found in coral-associated consortia [35].

The composition of stable consortia described for bioremediation purposes in the literature does not follow a common pattern, reflecting the diversity of compounds in nature [36]. In fact, it has been suggested that substrate type could be the main driver of the structure of microbial consortia developing during enrichment processes [37]. For instance, the bacterial consortium composed of *Pandoraea* sp. and *Microbacterium* sp. has demonstrated efficient degradation of phthalic acid esters and their derivatives [38]; similarly, a microbial consortium with the ability to degrade triphenyl phosphate efficiently is composed of *Pseudarthrobacter*, *Sphingopyxis*, *Methylobacterium*, and *Pseudomonas* [39]. On the other hand, an estrogen-degrading consortium composed of Proteobacteria and Bacteroidetes phyla has also been documented [40]. Therefore, given this context, the composition found in A21 may be suitable for growing in wastewaters.

For this reason, the general growth parameters of A21 were evaluated. pH was found to influence the growth of the A21 consortium, with optimal growth observed near neutral pH values. No growth was detected at pH 4.0 (Figure 2C). The absence of growth observed at this acidic pH may reflect the sensitivity of the cyanobacterial component, as it has been reported that the cytoplasmic acidification in cyanobacteria is associated with reduced growth rates [41]. Interestingly, the pH of the medium tended to stabilize around neutral values under all tested conditions (Figure 2D), likely due to the release of the metabolites liberated from the consortium to mitigate stress.

Additional studies on A21 have shown that this consortium lacks the ability to fix nitrogen, exhibits toxicity in the presence of urea, and cannot utilize tested sugars (glucose, sucrose, lactose, arabinose, maltose, fructose, galactose, and mannose) as substrates. Notably, certain *Rhizhobium* species lack *nif* genes, indicating an inability to perform nitrogen fixation [42]. A similar mechanism may apply to *Pseudorhizobium* strains within the consortium. Experimental data on A21 further support that an external nitrogen source is essential for its growth.

Furthermore, the inability of A21 to grow heterotrophically on the tested sugars could suggest two possible explanations: (i) the absence of a critical cofactor or vitamin in the medium may prevent bacterial growth within the consortium, or (ii) the *Cyanobium* component is incapable of heterotrophic growth on these sugars.

In recent years, biological filtration and bioremediation strategies for water reclamation have increasingly been based on microbial consortia, i.e., microbial communities formed by photosynthetic (cyanobacteria or microalgae) and heterotrophic (bacteria) microorganisms, living in a synergistic relationship, such the proposal for recycling wastewaters from household appliances [43]. The microbial combination found in A21, which includes two robust strains involved in degradation (*Sphingopyxis* and *Pseudorhizobium*) in combination with a cyanobacterium, represents a potential candidate for further investigation into its full range of capabilities.

First, the potential for growing in wastewaters was assessed and, for that, A21 consortium survival and growth was evaluated in different wastewater samples (Figure 3). A21 was able to growth in all wastewater samples tested (primary, secondary, or tertiary treatment effluents). However, it is noteworthy that the lower nutrient content in the tertiary-treated wastewater may have limited the growth of the consortium.

The consortium’s capability to thrive in wastewater depends on the mutualistic interaction of their microorganisms: heterotrophic bacteria can consume the organic matter, mineralizing nutrients and releasing CO_2_, while cyanobacteria utilize nutrients and CO_2_ and supply oxygen for the catalytic processes of the heterotrophic partners [44]. Consequently, bacteria and microalgae can facilitate the recycling of various compounds from wastewater (e.g., nitrogen or phosphorus and even organic matter) [10,45]. In this context, wastewater treatment using microalgae, coupled with biomass valorization, presents a promising sustainable and efficient approach [46]. However, maintaining axenic cultures under field conditions is not feasible, particularly when using wastewater, which introduces additional complexity. Therefore, the use of microbial consortia is challenging as they display greater robustness and could also accomplish complex tasks through division of labor [47]. Recent studies have suggested that consortia used in wastewater treatment can be economically viable [48]. In fact, although an initial native consortium evolved into two different consortia, both could still grow in the presence of wastewater while yielding interesting commercial bioactive compounds, such as the antioxidant and antimicrobial lecanoric acid and ω-3 and -6 fatty acids, which could be used as functional foods [48].

*Sphingopyxis* species are recognized for their extensive metabolic versatility, including the capacity to degrade xenobiotics and pollutants. This is due to the plethora of metabolic genes within its genome that encode enzymes involved in benzoate, phenylpropanoids, aminobenzoate, chlorinated compounds, microcystins, or aromatic compound degradation [27,49,50]. Some of the adaptive mechanisms described for *Sphingopyxis* that contribute to its tolerance to toxicants include alterations in the composition of bilayer-forming lipids and anionic lipids under stress conditions, which influences the metabolic activity and substrate uptake potential [51]. Although there is no direct evidence that this genus grows on steroids, its genomes contain some steroid catabolic genes (e.g., *Sphingopyxis alaskensis* RB2256 JGI project id: 3634495, https://www.genome.jp/brite/sal01000+Sala_1660, accessed on 20 November 2024), for instance, 3-ketosteroid-delta-1-dehydrogenase (NCIB::PHR19777.1); steroid 5-alpha reductase family enzyme (NCIB::SMQ77043); or steroid monooxygenase (NCIB::PHR16461.1), among others. It is remarkable that 3-ketosteroid-delta-1-dehydrogenase activity is involved in the AD/ADD conversion pathway [52]. 

In addition to Sphingopyxis, *Cyanobium* may also contribute to steroid metabolism. Although no studies to date have reported the growth of *Cyanobium* on steroids, there is some indirect evidence suggesting its involvement. For instance, when studying the effects of the xenoestrogen α-ethinylestradiol (EE), a compound commonly found in contraceptives and potentially released into the environment via sewage effluent, it was observed that *Cyanobium parvum* exhibited a bloom after exposure to EE in still-water microcosms [53]. Moreover, an examination of the *Cyanobium* genomes for steroid-related genes revealed the presence of several genes potentially involved in steroid catabolism, including 5α-reductases, also known as 3-oxo-5α-steroid 4-dehydrogenases (NCIB::PZV20400.1), and a GMC family oxidoreductase (NCIB::WP_106502785). In contrast, genomic analysis of *Pseudorhizobium* revealed no steroid-related genes, except for a ketosteroid isomerase-related protein (NCIB::WP_395515770.1) of unknown function.

Recently, a consortium based on *Chlamydomonas* has been proposed for the detoxification of wastewater within a circular economy framework. This approach leverages the organism’s potential for sustainable contaminant reduction while simultaneously facilitating resource recovery and valorization of microalgal biomass [54,55]. Among the possible applications of this biomass are its use as a biofertilizer, in biofuel production, and in the enhancement of hydrogen production. Similarly, the biomass from the A21 consortium following wastewater treatment could have various applications. However, a comprehensive study is needed to determine its most optimal uses. For example, it would be valuable to investigate whether A21 has genetic transformation capabilities, as this characteristic could open new avenues for genetic manipulation within the community context [56]. In fact, identifying the appropriate engineering tools is crucial for a deeper understanding of the microbial community and for maximizing its potential applications.

In this work, several attempts at genetic transformation were made by triparental mating using several pSEVA vectors displaying different origins of replication [57,58]. However, no successful transformation was achieved under the experimental conditions tested. Previous efforts to transform complex bacterial communities have been reported, such as the combination of environmental transformation sequencing (ET-seq), which utilizes non-targeted transposon insertions for mapping, and DNA-editing all-in-one RNA-guided CRISPR-Cas transposase (DART) systems, which enable targeted DNA insertion into organisms identified as amenable to transformation by ET-seq. These approaches have been applied to microbiota in soil and infant gut samples (citation). Given these developments, it may be valuable to apply such editing techniques to A21 in future experiments.

## 4. Materials and Methods

### 4.1. Origin and Isolation of Cyanobacteria-Based Consortia

A native cyanobacteria–bacteria consortium including the cyanobacterium *Cyanobium* sp. was isolated from wastewater samples from an EDAR, Madrid, Spain. All samples were subcultured in sterile conditions on BG11, BG13 [59,60,61], and UTEX medium on 1.5% agar plates or liquid medium at 30 °C ± 2 °C under 100 μE·m^−2^·s^−1^ of continuous white light, under orbital shaking (150 rpm). The BG11 (PhytotechLabs, St Lenexa, KS, USA) [62]. BG11 medium was buffered to pH 7.5 with 10 mM HEPES (Sigma-Aldrich, Madrid, Spain) and then supplemented with antifungals nystatin and cycloheximide both at a concentration of 100 μg/mL. Once the consortium was isolated from eukaryote contaminants, the presence of other bacterial strains was confirmed by growing the consortium on LB agar plates and on BG11 plates with 10 mM glucose, incubated at 30 °C on darkness.

Cultures of A21 were harvested at different growth phases and preparations of these cultures were photographed under a microscope (Leica, model DM750) (Wetzlar, Germany) and then processed by the LAS V4.2 software (Wetzlar, Germany). Other strains used in this work are shown in Table 2.

### 4.2. Consortium Composition

To understand the composition of the consortium, a metagenomic analysis was performed using the 16S rRNA amplicon sequencing protocol. Compendium samples were obtained from biomass from three different Petri plates harvested at one month of culture on BG11, scratching the surface of the culture lightly at different points along plates using a tip. Genomic DNA of wet biomass was extracted using the SpeedTools DNA Extraction Kit (Biotools, Madrid, Spain). The quality and quantity of the DNAs were determined by spectrophotometry, with the absorbance measurements at 260 nm, 280 nm, and the A_260_/A_280_ ratio.

Library construction was performed using a custom protocol. Briefly, amplification of the V1–V2 and V3–V4 regions of the prokaryotic 16S rRNA was carried out using primer sequences previously described in the literature [64,65], with an additional tail corresponding to the first part of the illumina adaptor (Appendix A). A first PCR was performed using 5 µL of isolated DNA and specific primers (final concentration 0.25 µM each) with the DNA AmpliTools Master Mix 2X (Biotools, Madrid, Spain) in a final reaction volume of 20 µL. The PCR program consisted of 30 cycles (95 °C for 30 s, 55 °C for 30 s, and 72 °C for 30 s). PCR products were purified using NucleoSpin Gel and PCR Clean-up (Macherey-Nagel) (Düren, Germany). One microliter of purified products was then used in a second PCR round, during which two different 8-nt sample-specific indexes, one on both sides of the amplicons, were incorporated along with the final region of the illumina adapter (Appendix A). This second PCR was performed with sample-specific primers (final concentration 0.25 µM each) using the DNA AmpliTools Master Mix 2X (Biotools, Madrid, Spain) in a final volume reaction of 20 µL. The PCR program was performed for 15 cycles (95 °C for 30 s, 55 °C for 30 s, and 72 °C for 30 s). The PCR products were visualized on a 1% agarose gel to corroborate the correct size of the amplicons. A negative control was processed in parallel for both regions analyzed. An equal volume (10 µL) of each PCR product was mixed, followed by Proteinase K (Thermo Scientific, Waltham, MA, USA) treatment before purification with SPRIselect beads (Beckman Coulter, Brea, CA, USA).

Nanopore sequencing libraries were constructed from 1 µg of purified PCR product using the ligation sequencing amplicons V14 kit (SQK-LSK114; Oxford Nanopore Technologies, Oxford, UK) according to the manufacturer’s instructions. Sequencing was performed in a MinION Mk1C device (Oxford Nanopore Technologies) using a FLO-MIN114 (R10.4.1) flow cell (Oxford Nanopore Technologies). Data acquisition and basecalling were performed using MinKNOW 23.07.12 with the fast basecalling model, retaining reads with a quality score greater than 8. FASTQ files containing accepted sequences were used as input for subsequent bioinformatics analysis. Briefly, the first part of the amplicons, corresponding to Illumina adapters, was searched across all sequences, and the results were used to build a hidden Markov model using the hp_bootstrap.py software (https://github.com/nanoporetech/hammerpede, accessed on 6 September 2024). The resulting model was applied in the Pychopper software (https://github.com/epi2me-labs/pychopper, accessed on 7 September 2024), where only sequences containing both adapters and in the correct configuration were retained for further analysis. Filtered sequences were demultiplexed according to the specific dual index introduced during library construction using the minibar.py software (https://github.com/calacademy-research/minibar, accessed on 7 September 2024). Primer regions were deleted from the sequences by trimming an additional 40 nt from the 5′ end and 30 nt from the 3′ end of the amplicons using the fastx_trimmer tool from the FASTX toolkit software (https://github.com/agordon/fastx_toolkit, accessed on 7 September 2024). A size filtering step was then applied, retaining sequences between 250 and 350 bp for the V1–V2 region and 350 and 500 bp for the V3–V4 region using SeqKit software, version 2.8.0 [66]. At this stage, the clean reads, grouped according to sample and fragment, were processed through the metagenomics workflow (version 2.8.0; Oxford Nanopore Technologies) implemented in the Epi2me Labs software (version 5.1.8.; Oxford Nanopore Technologies) (https://github.com/epi2me-labs/wf-metagenomics, accessed on 8 September 2024), analyzing the data using Minimap2 software (version 2.26-r1175) as the classifier and NCIBI_16s_18s as the reference database.

### 4.3. Consortia’s Growth Monitoring

To assess the growth of the consortium over a 15-day period in BG11 liquid media, optical density (OD) was measured daily using a spectrophotometer E-1000UV (Biogen, Madrid, Spain). Measurements were taken at a wavelength of 750 nm, which captures all microorganisms and minimizes interference from pigment absorption bands (350–700 nm) [67]. Each reading was conducted in triplicate to ensure accuracy and reliability.

A cell count was also conducted every day with a Neubauer counting chamber under the microscope (Leica, model DM750) according to the method described by Guillard and Sieracki [68].

### 4.4. Growth at Different Conditions

To characterize the growth of the consortium A21, it was inoculated into 20 mL of BG11 medium in 100 mL Erlenmeyer flasks to an initial optical density at 750 nm (OD_750nm_) of 0.05 and grown under different salinity conditions, pH, or nitrogen sources. To assess growth at different salt concentrations, BG11 medium was prepared containing 0.1, 0.25, 0.5, or 1 M of NaCl. The influence of pH on growth was explored in BG11 buffered to pH 4, 9, and 11 with 10 mM Tris adjusted to each pH. To examine the strains for growth on different nitrogen sources, BG11 was modified by replacing the 16 mM of NaNO_3_ with 16 mM of NH_4_Cl or urea. Tolerance to urea was determined by adding this compound to final concentrations of 8 and 16 mM of BG11. As a control, strains grown under routine conditions (BG11 pH 7.5, 30 °C, 150 rpm, and continuous light 100 μE·m^−2^·s^−1^) were used. Cell growth was monitored by measuring the OD_750nm_ for a 10-day period. In all the growth experiments, three biological replicates were performed. To define the relationship between cell densities per unit OD at 750 nm wavelength, a hemocytometer was used to count the cells.

First, to assess the A21 heterotrophic growth, BG11 agar plates were prepared at final concentrations of 10 mM with different carbon sources: glucose, sucrose, lactose, arabinose, maltose, fructose, galactose, and mannose. The tests were performed with spots of 10 μL at OD_750nm_ = 1 onto BG11 plates to reduce the possibility of contamination. Plates were incubated at 30 °C in darkness for 30 days. Furthermore, the photosynthesis inhibitor DCMU (3-(3,4-dichlorophenyl)-1,1-dimethylurea) was added for a final concentration of 10 μM to make sure that the observed growth was heterotrophic. Cell growth was evaluated by plate counting after the incubation period. As control, *C. thermophila* D14 xerotolerant strain was used [69].

All trials were conducted with three replicates. Growth curves were analyzed with the DASH program fitting growth curves to the parametric Logistic and Gompertz growth models [70] (https://dashing-growth-curves.ethz.ch/, accessed on 30 September 2024) to determine the growth parameters. An analysis of variance (ANOVA) was performed to determine if there were differences between the growth parameters. Validity was measured statistically through a one-way analysis of variance test, and a *p*-value of <0.05 was considered significant (Appendix A). In all tables and graphs, the means are reported with the confidence interval for a significance level of 95%. Further, Tukey’s test was applied for post hoc analysis when needed (Appendix A). All data were processed using Excel 10.0 for Windows.

### 4.5. Wastewater Growth

The behavior of A21 in wastewater from treatment plants in the Community of Madrid was analyzed. For the experiment, primary, secondary, and tertiary treatment wastewater from treatment plants in the Community of Madrid were used (Appendix A). Three bottles were prepared, one for each treatment, to which the corresponding residual water and distilled water were added in the same quantities (1:1 ratio). A final volume of 20 mL was used and with an initial inoculum of OD_750nm_ of 0.5 for A21. Growth was analyzed on days 5, 10, and 15 of the experiment. In all cases, three biological replicates were performed.

### 4.6. Analysis of Steroid Biotransformation by Thin Layer Chromatography (TLC)

For this experiment, the protocol described in Guevara et al. [71] for cyanobacteria was adapted. For a period of 25 to 30 days, strains A21, B7, and D14 were grown in liquid BG11 medium in a final volume of 20 mL with the different steroids. The sterols CHO and AD and the steroid hormone β-estradiol (E2), all from SIGMA-ALDRICH and prepared in cyclodextrins, were used at a final concentration of 0.1mM. Simultaneously, flasks of 20 mL of BG11 with each of the steroids, but without inoculum, were carried as controls. On certain days, OD_750nm_ measurements were taken from the flasks with inoculum and 500 μL aliquots were collected for freezing from all flasks.

To carry out steroid extraction, once the Eppendorf tubes were thawed, they were centrifuged for 3 min at 13,300 rpm and the supernatant (500 μL) was transferred to 50 mL Falcon tubes. From here, the extraction hood was used: 2 volumes of chloroform (1 mL) were added to each of the Falcons and vortexed for 30 s. They were subsequently frozen at −80 °C for 15 min. After this time, the frozen aqueous phase was carefully broken up and the chloroform phase was transferred to 2 mL Eppendorf tubes. For the chloroform to completely evaporate, the Eppendorfs were placed at 65 °C in a thermoblock.

Once the steroids were extracted, the biotransformation was analyzed using thin-layer chromatography (TLC). The samples were resuspended in 100 μL of chloroform and 2 μL of the sample, in the case of cholesterol it was 8 μL, and 1 μL of the standards (1 mg/mL) were loaded on silica gel plates (TLC Silica gel 60 F254, Merck Millipore, Burlington, MA, USA), trimmed so that the stroke was 4 cm and there was 0.5 cm separation between the samples. The mobile phase used was chloroform/ethanol (95:5, *v*/*v*) for the analysis of β-estradiol and hexane/ethyl acetate (10:4, *v*/*v*) for cholesterol and AD. The different steroids were revealed by immersing the plate in a methanol/sulfuric acid solution (90:10, *v*/*v*), which was then placed on a plate shaker at 230 °C.

### 4.7. Triparental Conjugation

#### 4.7.1. Antibiotic Sensitivity Test

To determine the sensitivity to different antibiotics of the isolate A21, 4 concentrations (μg/mL) of the following antibiotics were used (Appendix A): Kanamycin 25–100 (Km), Chloramphenicol 7.5–75 (Cm), Streptomycin 2–20 (Stm), Spectinomycin 2–20 (Spm), Neomycin 25–250 (Nm), Erythromycin 20–200 (Em), Gentamicin 2–20 (Gm), and Nalidixic Acid 7.5–75 (Nal). *Synechocystis* sp. PCC 6803 (SYN) was used as a control. SYN and A21 were placed as 20 μL droplets at an initial OD_750nm_ of 0.5. A control plate was carried out without any antibiotic and cultured at 30 °C and 100 μmol photon/m^2^s. After 15 days, it was assessed whether there was growth.

#### 4.7.2. Transformation Process

The conjugation protocol used was based on that described in Elhai and Wolk, 1998 [72] for spot conjugations, which allows several plasmids to be tested at the same time. Cyanobacterium *Synechocystis* sp. PCC 6803 was used as control. *E. coli* HB101 with the conjugative plasmid (pRK2013) was used in combination with one of the following options: (a) *E. coli* carrying the cargo plasmid of interest; (b) *E. coli* that, in addition to carrying the cargo plasmid of interest, had the helper plasmid (pRL623). The cargo plasmids used were of two types: on the one hand, plasmids pSEVA 221, 231, 241, and 251, which are interesting modular plasmids because they have different origins of replication and, on the other hand, plasmids pSL2680 and pSEVA251-Cpf1, which contain the CRISPR/Cpf1 (Cas12a) cassette, for genome editing. All plasmids used in triparental conjugation are listed in Appendix A.

For the procedure, the cyanobacteria were grown for 15 days in BG11 until they had sufficient biomass. In the case of *E. coli* HB101, they were grown the day before conjugation was carried out in liquid LB with the corresponding antibiotic: the *E. coli* with the conjugative plasmid (pRK2013) and those with each of the cargo plasmids, but without the helper plasmid, grew in LB medium with Km50; meanwhile, those that, in addition to the charge, had the helper plasmid grew in LB with Km50 and Cm34.

Spot conjugation was carried out completely under sterile conditions in three phases: (1) preparation of *E. coli*; (2) preparation of cyanobacteria; and (3) mixing of conjugation or mating. In the first and under the burner, the mixture of *E. coli* HB101 with the different plasmids was prepared in the following proportion: 1 mL of charge and 0.75 mL of conjugative, resuspended in a 60 μL final volume. Taking this and the number of conjugations to be carried out into account, 2 mL of *E. coli* culture containing the cargo plasmid and 1.5 mL of *E. coli* with the conjugative were centrifuged for 1 min at 13,300 rpm. They were then resuspended in 1 mL of LB without antibiotic, two washes were performed, and then they were resuspended in 500 μL of LB. Subsequently, the 500 μL of the strain with the conjugative plasmid was mixed with the 500 μL of each of the strains with the cargo plasmid, centrifuged again for 1 min at 13,300 rpm, and then a final wash was carried out in LB without antibiotic. Finally, it was resuspended to a minimum volume of 120 μL in LB and allowed to incubate at 37 °C for 1 h. During this time, the second phase of conjugation, the preparation of the cyanobacteria in the flow hood, was carried out.

Cyanobacteria were prepared at an OD_750nm_ of 20, 8, and 0.8 under sterile conditions. In the last phase, the conjugation was mixed on a sterile 47 mm Milipore HAF membrane on BG11 plates with 1.5% agar and 5% LB. An amount of 7 μL of each of the cyanobacteria preparations was mixed with 7 μL of each *E. coli*, in a 1:1 ratio. Of this mixture, 10 μL was deposited as a spot on the membrane. Additionally, 10 μL drops of cyanobacteria with the lowest OD_750nm_ (0.8) were placed on each of the plates as a negative control, without mixing it with the *E. coli*. The plates were left to incubate for 48 h at 30 °C and at 100 μmol photon/m^2^s. After this time, the membranes were changed to BG11 agar plates with the selection antibiotic, Km50, and incubated again under the same conditions. If the conjugation was successful, it would take a week to see transformants. For maintenance, every 7–10 days the membrane would be changed to a new plate with the selection antibiotic.

## 5. Conclusions

The metagenomic analysis of the A21 sample, isolated from wastewater after several streaks aimed at obtaining an axenic isolate, revealed that it constitutes a stable consortium primarily composed of *Cyanobium* (14.2%), *Sphingopyxis* (31.78%), known for its biodegradation potential, and *Pseudorhizobium* (15.68%), recognized for its toxification abilities. This consortium thrives under mesophilic conditions with specific pH and salinity requirements, necessitating a source of urea, and is capable of growing and degrading steroids. It is likely that this consortium was established in urban water environments, where emergent contaminants, such as steroids, are more prevalent than extreme pH or salinity conditions. Given the robustness of the consortium, A21 represents a promising alternative for the treatment of emerging contaminants, such as steroids, in aquatic environments.

## Figures and Tables

**Figure 1 ijms-25-13018-f001:**
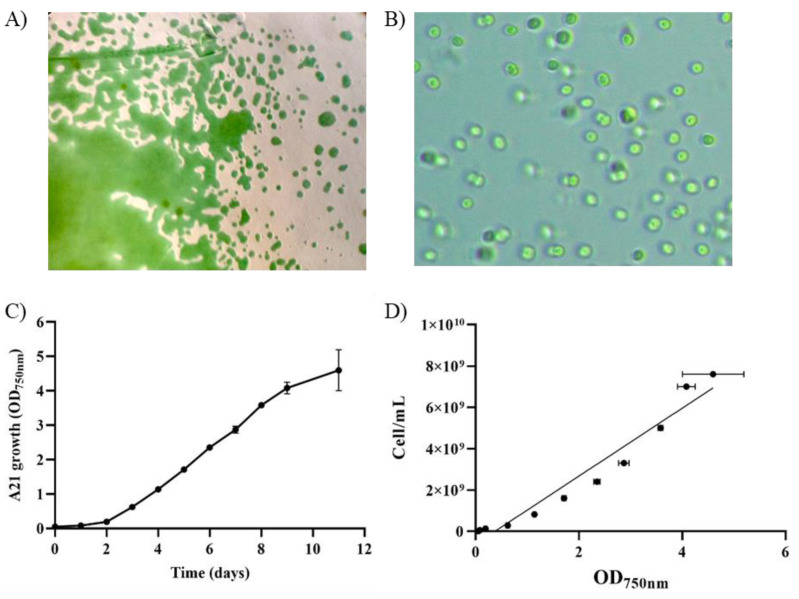
Isolation and characterization of A21, a cyano-based consortium. (**A**) Streak colonies (10× amplification); (**B**) bright-field photomicrographs of A21 consortium (100×). All pictures were taken from colonies or cultures growing on BG11; (**C**) growth curve on BG11. Average OD_750nm_ of three biological replicates together with the standard deviation is depicted; (**D**) relation between OD_750nm_ and cell count of A21 consortium, R^2^ = 95 (Y = 1642786425X − 611097024).

**Figure 2 ijms-25-13018-f002:**
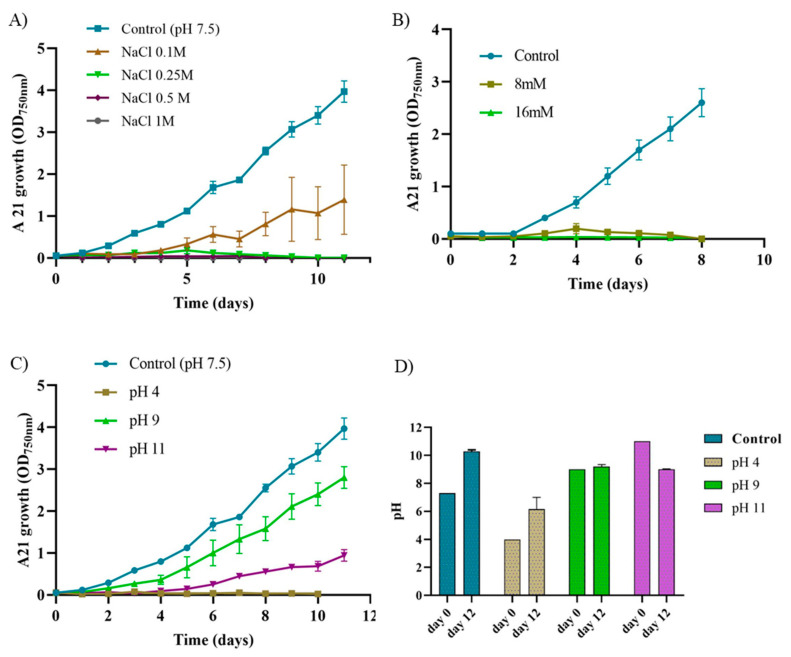
Growth characterization of consortium A21 under various conditions. (**A**) Effect of NaCl concentration on growth. BG11 medium growth served as control; (**B**) tolerance of A21 to urea. Urea was added to BG11 complete medium at concentrations of 8 mM and 16 mM. Growth under standard conditions (BG11 medium, pH 7.5, 30 °C, 150 rpm, and continuous light at 100 μE·m^−2^·s^−1^) was used as control; (**C**) effect of pH on A21 growth. BG11 medium buffered with 10 mM Tris at pH 4, pH 9, and pH 11 was used, with 10 mM HEPES at pH 7.5 serving as control; (**D**) pH levels before and after growth. In all cases, the graphs show the average OD_750nm_ of three biological replicates together with the standard deviation (n = 3).

**Figure 3 ijms-25-13018-f003:**
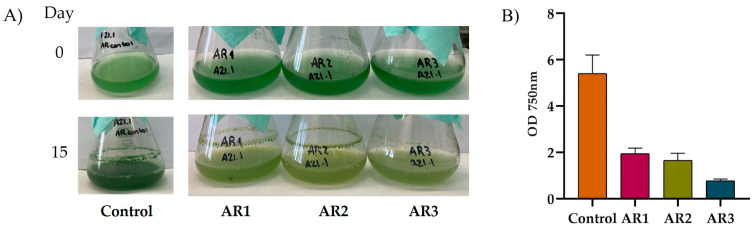
A21 growth in wastewater from treatment plants in the Community of Madrid. Photographs were taken on day 0 (initial OD_750nm_ of 0.5 for all the treatments) and day 15. (**A**) AR1: primary treatment water; AR2: secondary treatment water; and AR3: tertiary treatment water. (**B**) OD_750nm_ obtained after 15 days of growth. Data shown represent a representative experiment. Control: growth on BG11 medium.

**Figure 4 ijms-25-13018-f004:**
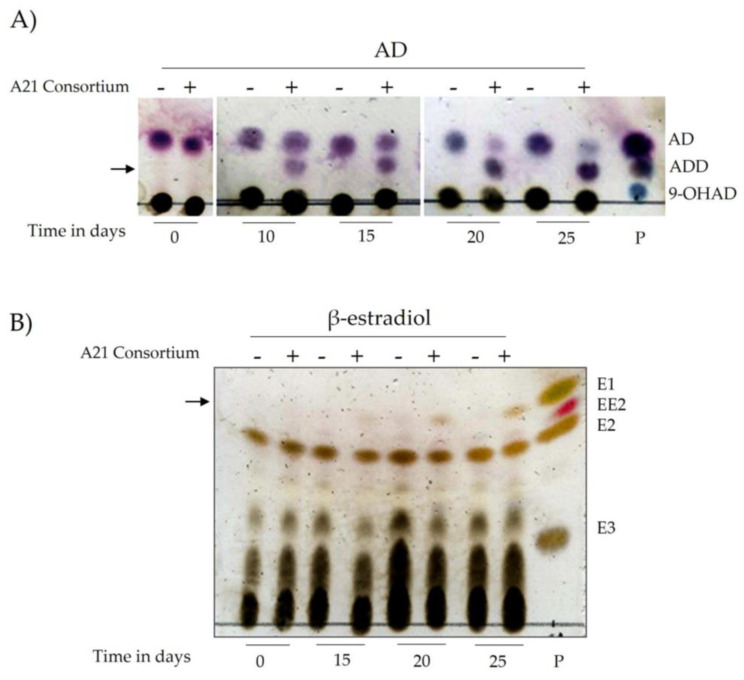
Steroid degradation by the A21 consortium. Thin-layer chromatography (TLC) analysis of steroid biotransformation during A21 consortium growth on BG11 medium supplemented with either 4-androsten-3,17-dione (**AD**) or β-estradiol at 0.1 mM over 25 days. (**A**) Degradation of AD and intermediate formation, including 1,4-androstadien-3,17-dione (**ADD**) and 9α-hydroxy-1,4-androstadien-3,17-dione (**9-OHAD**). (**B**) Transformation of β-estradiol (**E2**) into estrone (**E1**), estriol (**E3**), and 17α-ethinylestradiol (**EE2**). Controls include a sterile medium without the A21 consortium (-) and medium inoculated with the consortium (**+**). P: purified standards for reference. The arrow indicates the intermediate metabolites.

**Table 1 ijms-25-13018-t001:** Prokaryote taxa identified in the A21 cyanobacteria consortia and their relative abundance as a percentage of the total OTUs. Nineteen most abundant prokaryotes OTUs are listed.

Taxa				OTU Relative Abundance (%)	
Phylum	Class	Order	Family	Genus	%
Cyanobacteria	Cyanobacteria	Synechococcales	Prochlorococcaceae	*Cyanobium*	14.24
Proteobacteria	Alphaproteobacteria	Sphingomonadales	Sphingomonadaceae	*Sphingopyxis*	31.79
				*Sphingomonas*	9.97
				*Aquisediminimonas*	0.56
				*Hephaestia*	0.26
				*Sphingobium*	0.17
		Hyphomicrobiales	Boseaceae	*Bosea*	6.15
			Brucellaceae	*Brucella*	0.16
			Devosiaceae	*Devosia*	0.75
			Hyphomicrobiales	*Enhydrobacter*	1.39
				*Pseudorhodoplanes*	0.49
			Methylobacteriaceae	*Methylorubrum*	9.02
				*Microvirga*	0.15
			Phreatobacteraceae	*Phreatobacter*	0.18
			Phyllobacteriaceae	*Aquamicrobium*	0.23
				*Mesorhizobium*	1.46
				*Pseudaminobacter*	0.26
			Reyranellaceae	*Reyranella*	2.63
			Rhizobiaceae	*Pseudorhizobium*	15.68
				*Rhizobium*	2.14

**Table 2 ijms-25-13018-t002:** Strains used in this work.

Strain	Description	Reference
A21 consortium	Wastewater-isolated	This work
*Synechocystis* sp. PCC 6803 (SYN)	Cyanobacteria	[63]
*Escherichia coli* HB101	Conjugative strain	Promega Biotech Ibérica, SL, Spain

## Data Availability

The 18S rRNA data have been submitted to Genbank under accession number PP158241. The metagenomic data can be found at https://hdl.handle.net/20.500.14352/110855, accessed on 15 November 2024.

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
