# Peer review of "Characterizing A21: Natural Cyanobacteria-Based Consortium with Potential for Steroid Bioremediation in Wastewater Treatment"

_ijms, 2024, doi:10.3390/ijms252313018_

Round 1
Reviewer 1 Report (Previous Reviewer 1)
Comments and Suggestions for Authors
The authors' responses to my comments seem appropriate, and I accept the paper in its current version.

Reviewer 2 Report (Previous Reviewer 3)
Comments and Suggestions for Authors
The authors promptly responded to all of my inquiries.

This manuscript is a resubmission of an earlier submission. The following is a list of the peer review reports and author responses from that submission.
Round 1
Reviewer 1 Report
Comments and Suggestions for Authors
Dear Authors,
I believe the study has potential; the idea is good and could have significant biotechnological applications. However, under the current conditions, it cannot be published, primarily for three reasons: statistical errors are missing, the method used for steroid quantification, and whether the decontamination of steroids was actually done in wastewater or in BG11 medium. I think these are key points that need to be clarified
I think the introduction is well written.
Majors:
1-Have you tried to obtain axenic cultures of Cyanobium? Have you succeeded? Please discuss. Can Cyanobium axenica grow? Is the presence of the other two bacteria necessary to detoxify the steroids?
2-L164: “Triparental mating assays” I'm sorry, but I don't understand the purpose of these experiments. How important are they for your objective? Please clarify
3-Figure 3B does not have statistical error; I believe it is essential to show it.
4-Figure 4: I'm sorry, but in my opinion, the way you quantify steroid hormones using TLC is not sufficiently quantitative. You should use quantitative techniques such as HPLC. How many replicates have you done, and why not use HPLC? Please justify
-L273: “Other studies done on A21 have shown that this consortium cannot fix “nitrogen”, that the presence of urea was toxic and that the sugars tested were not a substrate for it.” The consortia between of microalgal and nitrogen-Fixing bacteria have been recently review, however you mention that your cyanobacterium does not fix nitrogen, could be due to the presence of nitrate in the BG-11 medium?. discuss
-Please discuss more elaborate on the biotechnological applications
-In which culture medium were the experiments of Figure 4 conducted? Please indicate it in the figure caption. It was wastewater?
-Please include a conclusions section
-L332: “plates and on BG11 plates with 10 mM glucose, incubated at 30ºC on darkness” Do you mean, then, that the consortium is able to grow in the dark? Discuss and elaborate on this better. How is it possible?
-L446: “flasks of 20 mL of BG11” Does this mean that the steroid degradation experiments were not done with wastewater but were conducted in BG11? If so, I don’t understand. What is the objective of the work? Wasn’t it the removal of steroids in wastewater? Please explain.
-Fig S1. So these data mean that Cyanobium can only grow well with nitrate, right? Is the Cyanobium genome sequenced? Does it have nitrate reductase? Could it grow with nitrate? If not, where does it obtain nitrogen? From the other two bacteria? Please discuss this in depth.
-As review recently, other strains of microalgae, such as Chlamydomonas in consortia, have been shown to bioremediate wastewater. What would this new consortium contribute compared to these already known ones? Please discuss
-What is the nitrogen composition in the three types of wastewater, and what implications does it have? Please discuss
Minors:
-L103: “end of the culture” Could you clarify better what this means? Please
-L172: “In order to see if this study if any” poor English please check
-L528: NH4Cl2. Typo
Reviewer 2 Report
Comments and Suggestions for Authors
I think the article entitled “Characterizing A21: Natural Cyanobacteria-Based Consortium with Potential for Steroid Bioremediation in Wastewater Treatment” needs serious adjustments.
For example, the abstract should be rewritten, the most significant results of the research should be in the abstract.
The article should be restructured, the order of the sections should be revised.
The introduction should address the problems of emerging contaminants, especially the steroids studied in this work. The problems of steroids, market, metabolic degradation pathways, effect on the environment, ecosystems, etc. should be mentioned.
For example, how is it possible that the materials and methods section is after the results and discussion sections. The quality of the figures is very poor. Please write in an impersonal format. Please improve the discussion of the results, compare with other previously published works.
Please, write the conclusion of the work
Reviewer 3 Report
Comments and Suggestions for Authors
The manuscript presents novel findings regarding the characterization and bioremediation potential of the A21 microbial consortium. However, there are several areas that require clarification and revision to enhance the quality of the manuscript. Overall, the manuscript is of high interest but needs some improvement before acceptance.
Include a functional analysis of the metagenomic data or reference studies that elucidate the role of the organisms identified in the consortium (e.g., Sphingopyxis, Pseudorhizobium) in bioremediation or wastewater treatment.
Provide more discussion on why A21 is unable to utilize sugars. This could be supported by additional experiments or by citing relevant literature.
Elaborate on the biochemical mechanisms behind steroid biotransformation. Are there known enzymes or pathways in cyanobacteria or Alphaproteobacteria that could explain this activity? Including enzyme assays or genomic annotations (e.g., presence of steroid-degrading genes) could significantly strengthen the study.
Explore alternative genetic manipulation methods or discuss the challenges in more detail, especially regarding the use of CRISPR systems in environmental consortia. Comparing these results to similar unsuccessful or successful attempts in literature would help contextualize the difficulties faced.
Include a discussion on the feasibility and scalability of using A21 in industrial wastewater treatment systems, comparing it with other known bioremediation consortia or strategies
Provide more detailed descriptions in the Materials and Methods section regarding the number of biological and technical replicates and the statistical tests used.
Increase the resolution of the TLC images and provide clearer labels for each lane. Ensure that all growth curves include error bars and statistical significance indicators where applicable.
Please, discuss the limitations of the genetic manipulation approach and offer suggestions for future research to overcome these challenges.
Comments on the Quality of English LanguageThe manuscript is well-written, but there are a few awkward phrasing and grammatical issues. Careful proofreading would help correct minor typographical and grammatical errors.